# Paramagnetic Properties of Carbon Films

Bagila A. Baitimbetova [1,2,*], Yuri A. Ryabikin [2], Bagdat A. Rakymetov [2], Danatbek O. Murzalinov [2], Dinara O. Kantarbaeva [2], Bahat Duamet [1], Elena A. Dmitriyeva [2], Abay S. Serikkanov [2] and Kassym Yelemessov [1]

[1] Department of Materials Science, Nanotechnology and Engineering Physics, Satbayev University, Almaty 050013, Kazakhstan; bahat62@yahoo.com (B.D.); k.yelemesov@satbayev.university (K.Y.)

[2] Institute of Physics and Technology, Satbayev University, Almaty 050032, Kazakhstan; yuar1939@mail.ru (Y.A.R.); b.rakymetov@sci.kz (B.A.R.); d.murzalinov@sci.kz (D.O.M.); d.kantarbayeva@sci.kz (D.O.K.); dmitriyeva@sci.kz (E.A.D.); a.serikkanov@sci.kz (A.S.S.)

\* Correspondence: baitim@physics.kz

**Abstract:** This research paper presents the results of obtaining carbon films on various substrates (quartz, mica, and silicon) at temperatures from 0 °C (initial) to 800 °C through plasma chemical vapor deposition. The carbon films obtained on various films were studied using the method of electron paramagnetic resonance (EPR). EPR measurements were carried out on twenty samples at a perpendicular and parallel arrangement of the sample plane concerning the orientation of the magnetic field. When measuring the resonance conditions by changing the magnetic field, an EPR signal appeared in all of the deposited samples. The paper presents a general view of the EPR spectrum in all of the samples, including the signal intensity, *g*-factor, line widths, and normalized signal intensity of the carbon films on various substrates at temperatures from 0 °C to 800 °C. Studies show that with an increase in temperature, the normalized intensity of the EPR signal line increases during the deposition of a carbon in all deposited substrates (quartz, mica, and silicon) using the method of plasma decomposition of a mixture of methane and hydrogen.

**Keywords:** electron paramagnetic resonance; carbon films; magnetic field; paramagnetic property; intensity of signal; various substrates

## 1. Introduction

Recently, the technology of obtaining thin carbon films on various substrates has been significantly developed.

The surface, structure, phase composition, and physical and chemical properties of thin carbon films directly depend on the method of their preparation. For the deposition of carbon films, various types of substrates are selected (quartz, glass, mica, copper, titanium, crystalline, and monocrystalline silicon).

Depending on the synthesis temperature, the localized carbon can present the pair in an amorphous and nanocrystalline state. The electron paramagnetic resonance (EPR) method is a powerful and sensitive instrument for studying carbon films.

The peculiarity of this method lies in the fact that it makes it possible to detect unpaired electrons even at their lower concentration in the materials under study. This method describes their energy states or localization without changing or modifying the samples. Electron paramagnetic resonance spectroscopy is large-scale used to study carbon nanomaterials and is the most direct method for studying carbon films in condensed materials.

The main direction of modern electron paramagnetic resonance spectroscopy is to increase the sensitivity and spectral resolution through extending the operating frequency of spectrometers. Increasing the operating frequency also allows the registration of the EPR spectra for systems with high spin states, including multi-particle paramagnetic complexes, significant fine structure splitting, and exchange splitting. Typically, it doesn't permit

the observation of EPR spectra at conventional frequencies. In this case, the possibility of lowering the operating temperature also plays a fundamental role. This allows—in addition to increasing the sensitivity—the order of the energy levels to be established. It also determines the signs of quality structure parameters and exchange interactions [1–9].

In research work [10], the spectra of the study of carbon films using the EPR method are presented. The carbon films were obtained through laser-plasma deposition using various types of graphite targets.

The influence of the degree of structural perfection of a graphite target and the thermal stimulation of the substrate on the structure of deposited carbon films using the laser-plasma method has been studied.

The EPR spectra for the investigated carbon films of single signals registered at room temperature were revealed. In both cases, the EPR signal practically doesn't saturate up to the microwave radiation power of 100 mW.

Therefore, the observed paramagnetic centers are primarily associated with defects of the non-particle character. The thin films deposited on a "cold" substrate using targets of different graphite are described by a broad ($\Delta H \sim 15$ G) single line with a g-factor of 2.0011.

For pyrolytic carbon graphite, the considered line is ($\Delta H = 10$ G) and the g-factor is 2.0022. This quantity is close to the g-factor of the free electron, which is typical for diamond structures. At the same time, the g-factor value increases for carbon films received using graphite targets.

In [11,12], the EPR spectra of graphite used as an ion source cathode were considered. It is a singlet, weakly asymmetric line with a width between the derivative extremums of 12–14 G and from a $g$-factor of 2.010.

This result is typical of polycrystalline graphite, which has an imperfect structure containing traps of $\pi$-electrons in the valence band [13]. For carbon in various states (black coals, soot, activated carbon, and amorphous a-C thin films obtained via different methods), the EPR spectra also represent singlet weakly asymmetric lines with a smaller width (3–4 G) and a different g-factor, equal to $2.0027 \pm 0.0002$ [14]. The EPR spectra of the carbon film obtained through the deposition of the products of the vacuum-arc erosion of graphite on a glass substrate in the anode chamber of the ion source have a complex shape. Computer modeling showed that this spectrum can be represented as a superposition of two lines: one is a broad line with $g = 2.0052 \pm 0.0002$ (width 20 G), the other is narrower (5 G) and has $g = 2.0030 \pm 0.0002$.

Wagoner [15] investigated the electron spin resonance in quite perfect single crystals of pure graphite and established that the resonance arises from mobile charge carriers. The temperature dependence of the g-factor in a graphite monocrystal at a 90–300 K temperature range were studied through X-band EPR [16]. This paper gives a generalization of the McClure equation for the $g = g(T)$ data description, which is based on the assumption that a (mixing coefficient of the p bond) is temperature-dependent.

The EPR spectrum of the carbon film presents a narrow singlet line with $g = 2.0034$. The EPR line in crystalline graphite is due to conduction electrons and described by a highly anisotropic $g$-factor.

According to [17], at room temperature, $g_{\parallel} = 2.0050$ and $g_{\perp} = 2.0026$, with $g_{\parallel}$ being highly dependent on the temperature and impurities. Therefore, the line with $g = 2.0052$ found in the article can be attributed to the presence of graphite microcrystallites with some averaged $g$-factor in the deposited film.

In this work, we studied the temperature dependency of the EPR spectra of thin carbon films deposited on quartz, mica, and silicon substrates on the pre-annealing temperature.

## 2. Materials and Methods

The carbon films were produced using the AX5200S-ECR machine from Seku Technotron Corp (Tokyo, Japan), equipped with a microwave emitter for plasma excitation. The process involved plasma decomposition of a mixture of methane ($CH_4$) and hydrogen

($H_2$), leading to the deposition of carbon on different substrates like quartz, mica, and silicon. This resulted in the formation of carbon films with a darkish color.

The growth of carbon films on quartz, mica, and silicon substrates was carried out in the SEKI AX5200S microwave plasma-enhanced chemical vapor deposition (PECVD) reactor (Tokyo, Japan), operating at a maximum plasma power of 1.5 kW. The substrates were placed in the PECVD reactor and then evacuated to approximately $2 \times 10^{-7}$ Torr using an external mechanical pump. The growth conditions were maintained with a plasma power of 540 W and a gas pressure of 16 Torr (approximately 2.13 kPa). The hydrogen flow rate was set at 80 standard cubic centimeters per minute (sccm), and the samples were pre-treated at 500 °C for 5 min before the carbon film growth process. During the growth process, methane ($CH_4$) was introduced with a flow rate of 20 sccm, and the carbon films were allowed to grow for 5 min while the growth temperature was varied from 0 °C to 800 °C. To study the $CH_4$ to $H_2$ flow rate ratio, growth was conducted at a fixed temperature of 800 °C, plasma power of 540 W, pressure of 16 Torr, and H2 flow rate of 80 sccm. The $CH_4$ flow rate was varied between 10 and 50 sccm (Figure 1).

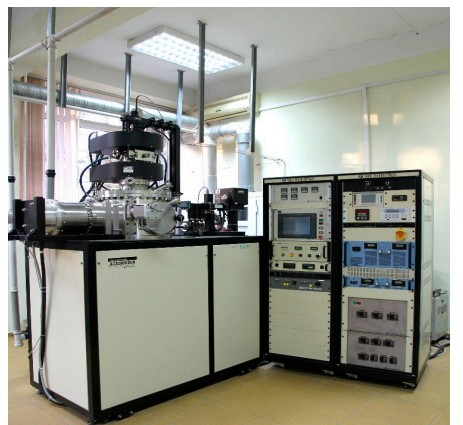 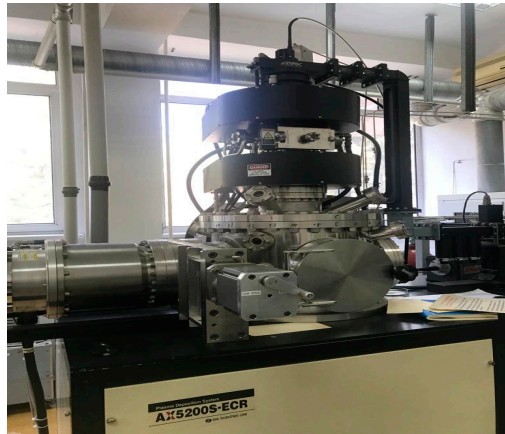

(**a**)

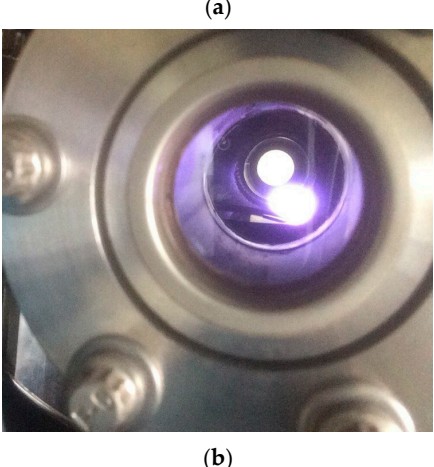

(**b**)

**Figure 1.** (**a**) General view of the equipment plasma-enhanced chemical vapor deposition PECVD and (**b**) View of the plasma in the reactor during the growth of carbon films.

The carbon deposition on the substrate to form the film required a time of 2 h. This process was performed using plasma excitation in a mixture of $CH_4$ + $H_2$ + Ag at a pressure of $3.8 \times 10^{-2}$ Torr. The input microwave power into the chamber was 540 W, and the reflected power was 84 W. The flow rate of $CH_4$ and $H_2$ into the chamber was equal at 20 cm$^3$/min.

The EPR measurements were conducted at room temperature, in an air atmosphere, using a spectrometer manufactured by the Japanese company, JOEL (Tokyo, Japan). The spec-

trometer operated in the 3 cm wavelength range and had a maximum sensitivity of $5 \times 10^9$ spins per sample with a 100 kHz magnetic field modulation. Measurements were carried out at room temperature at a frequency of 9450 MHz and a magnetic field of $336 \pm 10$ mT. The *g*-factor of the sample was determined using the known parameters of the manganese line. The temporary stability of the spectrometer is $10^{-6}$.

The resonator of the spectrometer is cylindrical. The sample is placed in the center of the resonator. A spectrometer feature is that the manganese sample is located outside the resonator, but under the influence of a scattered microwave field. This allows it to be used to calibrate the parameters of the recorded operating spectra. Its *g*-factor is stable when measuring the spectrum of a working sample.

EPR spectroscopy commonly utilizes the sample with secondary bivalent manganese ions within the lattice, frequently employing magnesium oxide (MgO) doped with $Mn^{2+}$ as the reference sample.

The signals of the carbon films were recorded with a g-factor approximately equal to 2, falling between the 3rd and 4th components of the 6 linear spectra from $Mn^{2+}$. The carbon film, measuring $3 \times 5$ mm in diameter, was positioned inside a specific glass ampoule that did not produce an EPR signal.

The EPR spectrum of this sample consists of 6 EPR lines between the 3rd and 4th components, which are usually recorded spectra of the studied sample. The ESR analysis of the third line, along with all other lines in the divalent manganese spectrum, indicates that there are no curves present when connecting its maximum and minimum points. The EPR spectrum of this particular sample exhibits 6 EPR lines positioned between the 3rd and 4th components, which are commonly observed in the spectra of the studied samples. The analysis of the third line, as well as all other lines of the divalent manganese spectrum, reveals that there are no curves along the line connecting its maximum and minimum points. This line appears to be completely straight.

When the resonance conditions were satisfied by varying the magnetic field within specific limits, the EPR signal became evident. This signal was then detected and transmitted to the printer to print the spectrum.

One significant advantage of EPR is that the provided data values of the *g*-factor of manganese enable the determination of experimental data for the line between these two manganese components. The g-factor plays a crucial role in the resonance condition $\Delta E = g\mu B \beta H$ (where $\mu B \beta$ is the Bohr Magneton), and it dictates the position of the line in the EPR spectrum. By knowing the *g*-factor, researchers can precisely analyze and interpret the EPR spectra to obtain valuable information about the sample's properties and electronic structure.

A free electron lacks an orbital momentum, and its *g*-factor is approximately equal to 2.0023. The *g*-factors of ions in the S-state are also close to this value. Even slight variations in this *g*-factor value (measured with an accuracy of 4–5 decimal places) signify alterations in the local environment of the center being studied. These changes can provide valuable insights into the electron's surroundings and help researchers understand the electronic structure and properties of the system under investigation.

EPR measurement was carried out on 20 sprayed samples at temperatures ranging between 0 °C (initial) and 800 °C. Overall, these measurements were 20 samples at both arrangements of the plane of the sample relative to the orientation of the magnetic field.

### 3. Results

Figure 2 shows the raw data of the carbon film on the substrate of mica at the temperature of 400 °C. Using this approach for measurement, the sample under study is positioned inside the resonator, while the manganese in the sample is situated outside the main volume, within a scattered Super High Frequency field. The phase of the signal is affected by the Super High Frequency (SHF) field in the resonator itself. This influence results in a distinct phase difference between the two signals. This influence results in a distinct phase difference between the two signals. It is known qualitative methods offer an effective way

of measurement of samples with $Mn^{2+}$ in MgO were selected for its reliability and validity. As a result, the signals from the sample and from the manganese have opposite phases. In this spectrometer, the sample of $Mn^{2+}$ in MgO is in the scattered microwave field outside the resonator; therefore, the phases of the investigated sample in the resonator and the sample of $Mn^{2+}$ in MgO are opposite. The EPR spectrum was calibrated according to a well-known standard, according to the works [18,19].

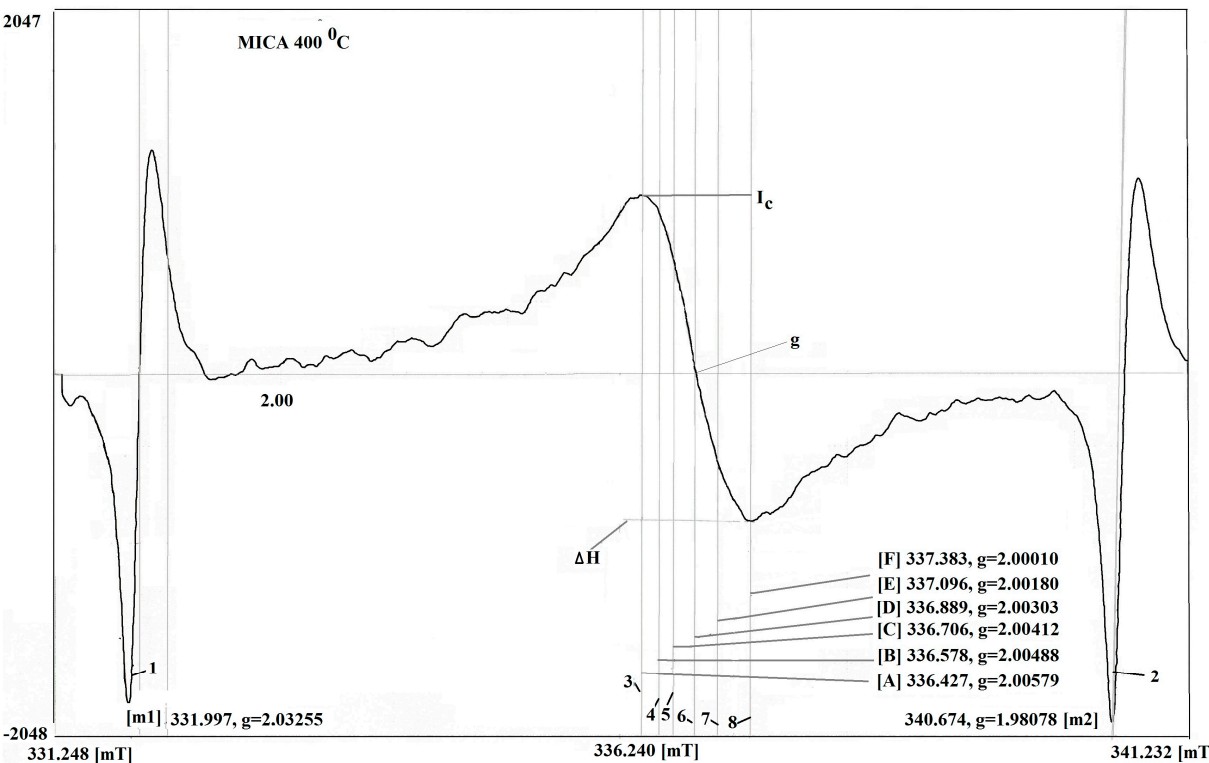

**Figure 2.** The raw data of carbon film on the substrate of mica at temperature 400 °C. The measurement of the perpendicular arrangement of the plane of sample relative to the orientation of the magnetic field 1–2—signal of Mn, 3–8—signals of sample and EPR characteristics, 6—signal of center.

Figure 2 shows the relation between the values determined by areas of 3:5 and 8:4 for the carbon film on the substrate of Mica at the temperature of sputtering 400 °C. It can be assumed that the thin carbon film is formed due to three components.

The characteristic parameters of the EPR spectra of the carbon film on the mica substrate are as follows: the caption for the picture presents the magnitude of the area, which has *g* = 2.00412–2.00420; this is typical of the EPR spectrum of graphene and its compounds. The second area is the g-factor equal to *g* = 2.0031. Such a *g*-factor of carbon components can be caused by carbon nanotubes.

The *g*-factor for the third component of the carbon film varies depending on the angle of the rotation of the sample in the magnetic field. Specifically, it ranges between 2.00118 and 2.00164. This variation in the *g*-factor suggests that the carbon film's electronic properties are influenced by its orientation relative to the magnetic field. This indicates the presence of the carbon film consisting of the various forms of graphite with some degree of crystallites [20,21].

The results obtained from measuring the EPR spectra of these samples are depicted in Figures 3–5, and detailed data are provided in Tables 1 and 2.

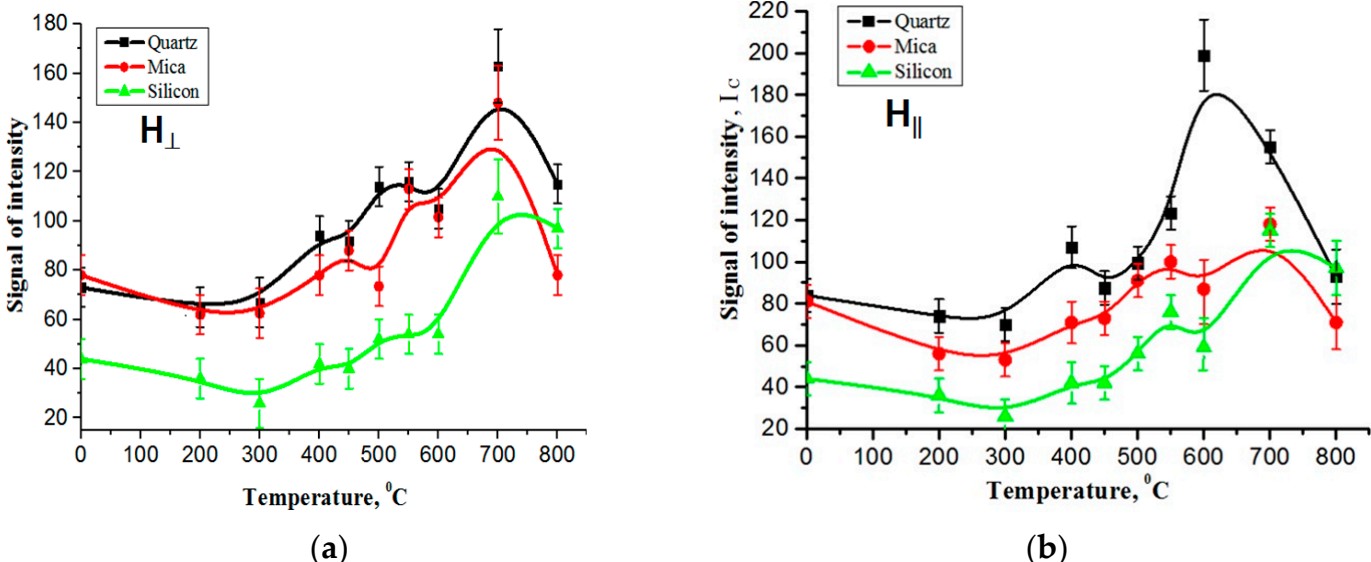

**Figure 3.** The EPR signal intensities of carbon films on various substrates exhibit temperature dependence: (**a**) the measurements were taken with the sample plane positioned perpendicular to the orientation of the magnetic field, (**b**) the measurements were conducted with the sample plane parallel to the orientation of the magnetic field.

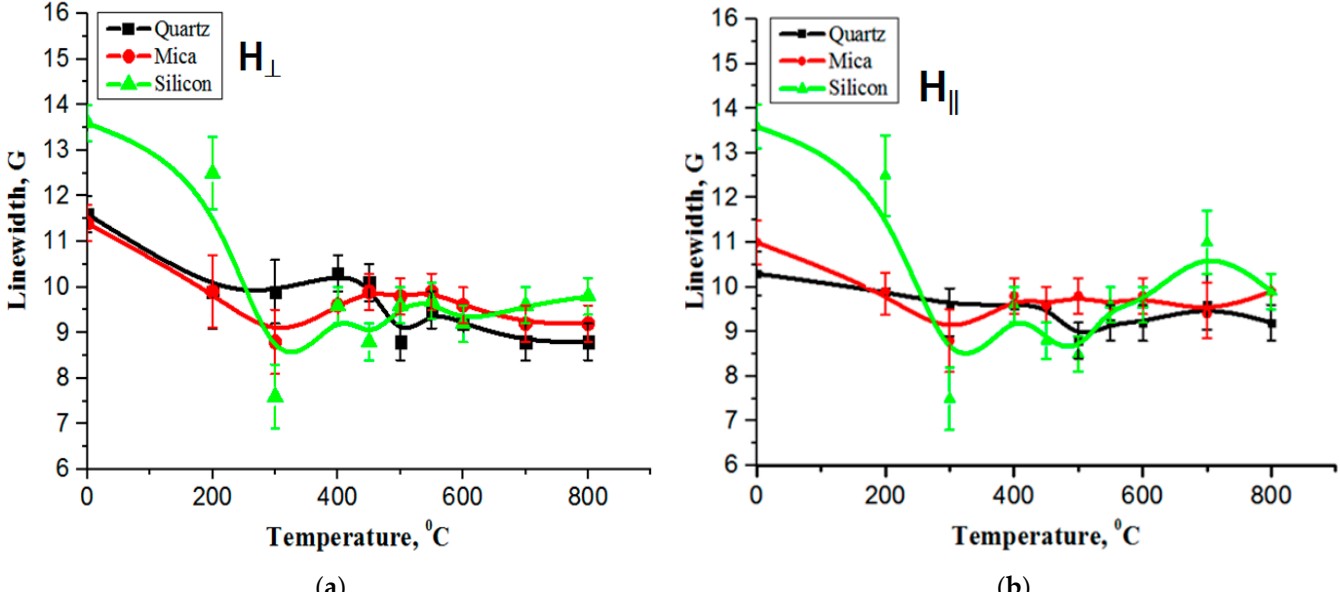

**Figure 4.** The temperature dependencies of the EPR line width for the carbon film on quartz, mica, and silicon substrates: (**a**) the measurements were taken with the sample plane positioned perpendicular to the orientation of the magnetic field, (**b**) the measurements were conducted with the sample plane parallel to the orientation of the magnetic field.

**Table 1.** The parameters of the EPR spectra obtained from the carbon film deposited on a mica substrate.

| No. | H, mT | *g*-Factor | No | H, mT | *g*-Factor |
|---|---|---|---|---|---|
| 1 | 331.997 | 2.03255 | 5 | 336.706 | 2.00412 |
| 2 | 340.674 | 1.98078 | 6 | 336.889 | 2.00303 |
| 3 | 336.427 | 2.00579 | 7 | 337.096 | 2.00180 |
| 4 | 336.578 | 2.00488 | 8 | 337.383 | 2.00010 |

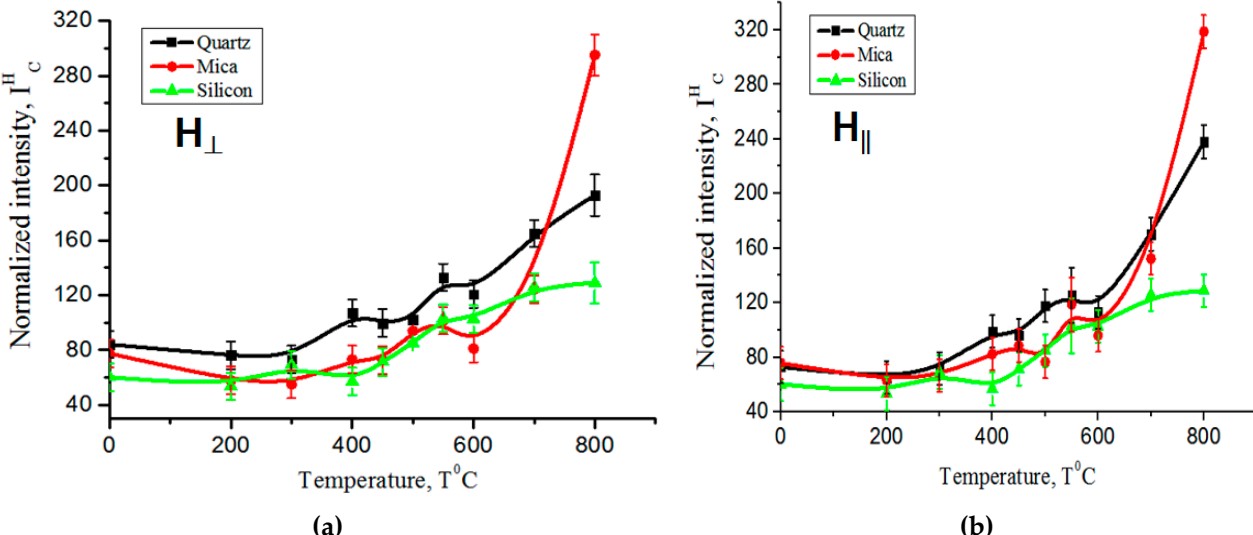

**Figure 5.** The dependence of the normalized EPR intensity on the temperature of carbon films on different substrates: (**a**) the measurements were taken with the sample plane positioned perpendicular to the orientation of the magnetic field, (**b**) the measurements were conducted with the sample plane parallel to the orientation of the magnetic field.

**Table 2.** Parameters of the *g*-factor at different arrangement of sample of carbon films on various substrates.

| TT, °C | $g_{\parallel}$-Factor of Spectra | | | $g_{\perp}$-Factor of Spectra | | |
|---|---|---|---|---|---|---|
| | Quartz | Mica | Silicon | Quartz | Mica | Silicon |
| 0 | 2.00283 | 2.00279 | 2.00264 | 2.00287 | 2.00293 | 2.00264 |
| 200 | 2.00292 | 2.00293 | 2.00249 | 2.00291 | 2.00319 | 2.00249 |
| 300 | 2.00304 | 2.00297 | 2.00221 | 2.00301 | 2.00293 | 2.00221 |
| 400 | 2.00324 | 2.00303 | 2.00257 | 2.00292 | 2.00288 | 2.00257 |
| 450 | 2.00311 | 2.00292 | 2.00278 | 2.00303 | 2.00238 | 2.00225 |
| 500 | 2.00302 | 2.00293 | 2.00294 | 2.00302 | 2.00303 | 2.00237 |
| 550 | 2.00326 | 2.00313 | 2.00281 | 2.00318 | 2.00313 | 2.00364 |
| 600 | 2.00319 | 2.00306 | 2.00304 | 2.00292 | 2.00326 | 2.00304 |
| 700 | 2.00329 | 2.00332 | 2.00303 | 2.00304 | 2.00322 | 2.00306 |
| 800 | 2.00327 | 2.00301 | 2.00303 | 2.00327 | 2.00292 | 2.00303 |

Table 1 presents the parameters of the EPR spectra obtained from the carbon film deposited on the mica substrate. These parameters may include various aspects of the EPR signals—such as the *g*-factors and resonance fields—used to analyze and characterize the carbon film in the experimental setup.

The measurement of the EPR spectra was conducted with the sample positioned perpendicular to the orientation of the magnetic field. In this arrangement, the plane of the sample is at a right angle to the direction of the magnetic field, which allows for insights into the properties and behavior of the sample under investigation. These spectra have significant implications for understanding interpreting all of the spectra of the paramagnetic features measured using EPR.

Figure 3a,b depicts the temperature dependencies of the EPR signal intensities for the carbon film on quartz, mica, and silicon substrates. The measurements were taken with the sample plane oriented both perpendicular and parallel to the direction of the magnetic field. These figures provide information about how the EPR signal intensities change with the temperature for different substrate materials and the orientations of the sample relative to the magnetic field. These data can be used for understanding the behavior of the carbon film under varying. The figure shows that when the temperature changes from room

temperature to 250–300 °C, the amplitude of the EPR signal decreases to an obvious extent, and then increases with the increasing temperature for all of the samples.

The temperature dependences of the intensity of the EPR signal of the carbon film on different substrates show that with the increasing temperature, the signal grows non-uniformly with the formation of intermediate maxima in both orientations of the sample plane relative to the orientation of the magnetic field.

Figure 3a,b shows that the EPR signal has one characteristic temperature region. The growth of the EPR signal with the temperature in the studied samples was observed in the interval $T$ = 400–700 °C when measured in the parallel position of the sample plane relative to the direction of the magnetic field (Figure 3b). In both orientations of the measurements at temperatures above 750 °C, they are transformed in one EPR line, which significantly decreases at 800 °C.

It was found that the temperature behavior of the EPR signal is nearly similar for both the perpendicular and parallel arrangements of the sample plane relative to the orientation of the magnetic field. In other words, the EPR signal intensities of the carbon films on different substrates exhibit comparable changes with the temperature, regardless of whether the sample plane is oriented perpendicular or parallel to the magnetic field. This observation suggests that the temperature-dependent properties of the carbon films are largely unaffected by the orientation of the sample plane in relation to the magnetic field. This means that the EPR signal refers to the same paramagnetic center, characterized by $g$ = 2.00322–2.00323 at $T$ = 750 °C. Generally, carbon films have a rather complex structural composition. It mainly includes graphene-like formations, nanotubes of various parameters, graphite of various organizations, and their oxides.

Figure 4a,b illustrates the temperature dependencies of the EPR line width for the carbon film on quartz, mica, and silicon substrates. The measurements were taken with the sample plane oriented both perpendicular and parallel to the direction of the magnetic field. It is known [22] that the EPR line width strongly depends on the microwave power, temperature, and amplitude of the magnetic field modulation. As the temperature increases, the line width of the carbon film on quartz and mica does not change.

This means that the effect of the Super High-Frequency power and the magnetic field modulation amplitude on the behavior of the EPR signals appears to be the same. This indicates that the centers have identical relaxation times, which may be related to the same local environment of the centers or the same dynamic properties [21–23]. In addition, the temperature conditions of the registration change the properties of the EPR spectra.

At the same time, the width of the carbon film lines from silicon decreases at 200–300 °C, and this has not yet been explained. A high concentration of paramagnetic centers in the carbon film results in temperature effects, indicating significant carbon film deposition. This high concentration of paramagnetic centers leads to the formation of a high level of carbon film, and such a film exhibits a strong interaction between its polarons due to their abundant presence. There is also a high probability of transformation between the polarons and bipolarons [10], which can greatly affect the relaxation of the spins (polarons) and their surroundings.

Table 2 presents the temperature dependencies of the $g$-factor for carbon films on various substrates when the sample plane is oriented both perpendicular and parallel to the direction of the magnetic field. The values in this table indicate how the $g$-factor of the carbon films changes with the temperature under these different experimental conditions.

When measured perpendicular and parallel, the magnetic field of the film from the silicon has a jump-like appearance.

This means that the unpaired electron is localized on a non-carbon atom and deflects at temperatures of 300 °C (($g_\perp \approx 2.00249$ and $g_\parallel \approx 2.00221$), 450 °C ($g_\perp \approx 2.00225$ and $g_\parallel \approx 2.00221$), and 550 °C ($g_\parallel \approx 2.00281$).

These deviations of the $g$-factors ($g_\parallel$ and $g_\perp$) in both orientations of the measurement strongly depend on the impurities of the samples [15].

The *g*-factor increases with the increasing temperature when the sample plane is measured parallel to the magnetic field of the carbon film from quartz and mica compared to the carbon film from silicon. This means that the organic radicals are practically isotropic at their insignificant concentration and differ insignificantly [23,24].

When measured perpendicular to the location of the sample plane relative to the magnetic field of the carbon film on mica at 450 °C, the *g*-factor ($g_\perp \approx 2.00238$) decreases. At the same time, at 550 °C, the *g*-factor ($g_\perp \approx 2.00364$) shows an increase.

This circumstance allows us to conclude that fluctuations in the g-factor value are not random, but are a manifestation of changes in the environment of an average radical (a broken **C-C** bond). This means that either local magnetization or local demagnetization occurs when the sample is exposed to an external constant magnetic field and the magnetic component of microwave radiation in the resonator [24,25].

Figure 5a,b depicts the normalized intensity of the EPR line as a function of the increasing temperature. The data in these figures demonstrate a clear, monotonic increase in the intensity of the EPR line as the temperature rises. This observation indicates a direct correlation between the temperature and the strength of the EPR signal. The change in the shape of the EPR signal line with the increasing temperature indicates the deposition of carbon on various substrates (quartz, mica, and silicon) during the plasma decomposition of the mixture of methane and hydrogen.

## 4. Conclusions

This research work investigated the dependencies of various EPR signal parameters of carbon films on different substrates. The studied parameters included the EPR signal intensity, line widths, *g*-factor, and normalized EPR line intensity. The measurements were taken with the sample plane oriented both perpendicular and parallel to the direction of the magnetic field.

The temperature dependencies of the EPR signal intensities of the carbon film on various substrates reveal an interesting behavior. As the temperature increases, the EPR signal exhibits non-uniform growth with the emergence of intermediate maxima. These intermediate maxima are observed in both orientations of the sample plane relative to the orientation of the magnetic field, i.e., both when the sample plane is perpendicular and parallel to the magnetic field.

In both orientations of the measurement at temperatures above 750 °C, they are transformed into a single EPR line. The line width of the carbon film on quartz and mica does not change with the increasing temperature. The results show that the centers have the same relaxation times, which may be due to the same local environment of the center, or to the same dynamic properties of the centers.

The data reveal that as the temperature increases up to 750 °C, the normalized intensity of the EPR signal amplitude exhibits a consistent and monotonic increase in all samples during the process of carbon deposition on various substrates. Whether on quartz, mica, or silicon substrates, the EPR signal intensity grows in a uniform and continuous manner with the rising temperature. This observation points to a direct relationship between the temperature and the strength of the EPR signals, indicating a systematic enhancement of the EPR responses as the temperature is elevated.

**Author Contributions:** Conceptualization, B.A.B. and Y.A.R.; software, B.A.B.; validation and formal analysis, Y.A.R. and A.S.S.; investigation D.O.M. and B.A.R.; writing—original draft preparation, B.A.B. and Y.A.R.; writing—review and editing D.O.K., E.A.D. and B.D.; visualization, B.A.R. and K.Y.; supervision, B.A.R. All authors have read and agreed to the published version of the manuscript.

**Funding:** The research presented in this work was supported by a grant from the Ministry of Education of the Republic of Kazakhstan, specifically the grant BR18574141 titled "Comprehensive multi-purpose program for improving energy efficiency and resource saving in the energy sector and mechanical engineering for the industry of Kazakhstan".

**Data Availability Statement:** Not applicable.

**Conflicts of Interest:** The authors declare no conflict of interest.

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
