# Peer review of "Paramagnetic Properties of Carbon Films"

_coatings, doi:10.3390/coatings13091484_

Round 1

Reviewer 1 Report

In this manuscriptthe authors deposited carbon films on various substrate including quartz, mica and silicon using plasma decomposition method of a mixture of CHand H2. According to the results of this research, the dependences of the parameters of the EPR signal of the carbon film are studied. Although the purpose of the study is interesting, it is more like a technical report than a research paper. I think that the following corrections should be made before the article can be published.

-       The presence of films on substrates should also be demonstrated by other chemical and/or physical characterization techniques.

-       In general, there are serious problems with the presentation of the article:

Both the abbreviations ESR and EPR are used for electron paramagnetic resonance. The second page is blank, etc.

-       Details about the system and how the plasma method is used should be added.

Author Response

Point 1: In this manuscript, the authors deposited carbon films on various substrate including quartz, mica and silicon using plasma decomposition method of a mixture of CHand H2. According to the results of this research, the dependences of the parameters of the EPR signal of the carbon film are studied. Although the purpose of the study is interesting, it is more like a technical report than a research paper. I think that the following corrections should be made before the article can be published. The presence of films on substrates should also be demonstrated by other chemical and/or physical characterization techniques.

Response 1:

The purpose of this work is to study the study of carbon films deposited on substrates of quartz, mica, and silicon by electron paramagnetic resonance (EPR). In this paper the results of electron paramagnetic resonance spectroscopy were used.

Point 2: Both the abbreviations ESR and EPR are used for electron paramagnetic resonance. The second page is blank, etc..

Response 2:

The samples were examined on an electron paramagnetic resonance (EPR) facility. This was a technical error. The observation has been corrected. Although in the abstract it was stated. This paper presents the results of an electron paramagnetic resonance (EPR) study of carbon films deposited on quartz, mica, and silicon substrates.

Abstract: This paper presents the results of an electron paramagnetic resonance (EPR) study of carbon films deposited on quartz, mica, and silicon substrates.

Introduction

Recently, there has been considerable interest in the development of technologies for depositing thin carbon films on various materials.

The morphology, structure, phase composition, and physical properties of thin films directly depend on the method of their production. Various types of substrates are selected for obtaining such films, such as: quartz, glass, crystalline and monocrystalline silicon, etc.

 Depending on the synthesis temperature, the localized carbon can be presented both in an amorphous state and in a nanocrystalline state. The electron paramagnetic resonance (EPR) method is a powerful tool for studying carbon nanomaterials.

Point 2:

  Details about the system and how the plasma method is used should be added.

Response 3:

Response 3:

Carbon films deposited on substrates of quartz, mica, and silicon was grown by using SEKI AX5200S microwave PECVD reactor with maximum plasma power 1.5 Kwatt. The samples with catalyst were loaded into the PECVD reactor, which was then evacuated to about 2 x10-7 Torr by an external mechanical pump. The growth conditions were fixed as fellow plasma power 540 watt and gas pressure 16 Torr (≈ 2.13 kPa). H2 flow rate 80 sccm (standard cubic centimeter per minute) and the samples were pretreated at 500 ºC for five minutes before the growth (in order to obtain carbon films). After the pretreatment, methane with flow rate 20 sccm was opened and the growth time was 5 minutes where the growth temperature was varied from 100 up to 800 ºC. For measuring the effect of the methane to hydrogen flow rate ratio, the growth temperature was fixed at 650 ºC, plasma power 540 watt, pressure 16 Torr , hydrogen flow rate 80 sccm and the growth time 5 minutes where the methane flow rate was changed from 10 up to 50 sccm (Figure 1).

а)

b)

Figure 1. а) General view of the equipment plasma-enhanced chemical vapor deposition (PECVD) and

  1. b) View of the plasma in the reactor during the growth of carbon films

Reviewer 2 Report

Results of this work clearly presented in tables and graphs. However, there is much need to improve grammar/text. Examples are:

line 130 date carbon

line 143 Thus suggests

Figure 2 y-xis labelled as Signal of Intensity

line 226 g-factor growths in comparison 

line 251 shows t nomalized

The results and conclusions address temperature dependences but the study was done over different substrates. I may have missed it but I didnt see and conclusion about comparisons of substrates. Even if there may be no significant effects of substrate it should be stated in results and conclusions. Same goes for orientation effects on magnetic field.

See above for comments on English language.

Author Response

Response to Reviewer 2 Comments

Point 1: Results of this work clearly presented in tables and graphs. However, there is much need to improve grammar/text. Examples are:

line 130 date carbon

line 143 Thus suggests

Figure 2 y-xis labelled as Signal of Intensity

line 226 g-factor growths in comparison 

line 251 shows t nomalized

Response 1:

line 130 date carbon. The observation has been corrected. Figure 1 shows raw data of the carbon film on substrate of mica  at temperature 400 0C

line 143 Thus suggests. The observation has been corrected. It can be assumed that the formed thin carbon film is due to three components.

Figure 2 y-xis labelled as Signal of Intensity- The observation has been corrected.

(a)

(b)

Figure 2. Dependence of EPR Signal Intensities on the temperature of carbon films on different substrates: a) at the perpendicular location of the sample plane relative to the orientation of the magnetic field, b) at the parallel location of the sample plane relative to the direction of the magnetic field

line 226 g-factor growths in comparison.  The g-factor increases with increasing temperature when the sample plane is measured parallel to the magnetic field of the carbon film from quartz and mica compared to the carbon film from silicon.

line 251 shows t nomalized. The observation has been corrected. Figure 4a,b shows that the normalized intensity of the EPR line increases monotonically with increasing temperature.

Point 2: The results and conclusions address temperature dependences but the study was done over different substrates. I may have missed it but I didnt see and conclusion about comparisons of substrates. Even if there may be no significant effects of substrate it should be stated in results and conclusions. Same goes for orientation effects on magnetic field.

Response 2: The effect of magnetic field orientation on the perpendicular and parallel arrangement of the plane of the carbon film on different substrates (glass, mica, and silicon) is the same. Obtaining the carbon film under the same conditions the influence of different substrates was no significant.

Reviewer 3 Report

Following are the key points reported in this work.

1. The investigation focused on studying the dependencies of EPR (Electron Paramagnetic Resonance) signal parameters of a carbon film on different substrates.

2. The temperature dependence of the EPR signal intensity of the carbon film on different substrates was analyzed. As the temperature increased, the EPR signal exhibited non-uniform growth with the formation of intermediate maxima, regardless of the orientation of the sample plane relative to the magnetic field.

3. At temperatures above 750°C, the intensities of the EPR lines were significantly reduced in both orientations of measurements.

4. The width of the carbon film line on quartz and mica substrates remained unchanged with increasing temperature.

5. The results indicated that the EPR centers had the same relaxation times, suggesting a common local environment or identical dynamic properties.

6. The normalized amplitude intensity of the EPR signal increased monotonically in all samples during carbon deposition on various substrates (quartz, mica, and silicon) as the temperature rose up to 750°C.

There are many mistakes, 

1. Electron paramagnetic resonance is abbreviated as ESR

2. EPR used in the introduction before abbreviating it

3. space need to be added in front of units in many places of the manuscript

4. English is not acceptable

5. same text repeated, text from lines 66-71 is same as text in lines 72-78

6. Line 130, it should be "raw data" in the place "raw date"

7. units indicated improperly in many places

8. Line, 251, "Figure 4a,b shows t normalized...." it should be replaced with "Figure 4a,b shows the normalized...."

The English employed in this manuscript does not meet the standards expected for a scientific journal.

Author Response

Response to Reviewer 4 Comments

Point 1: The objective of the study presented by the authors was to measure and determine the temperature dependence of the EPR signal in thin carbon films deposited on quartz, mica, and silicon substrates at different annealing temperatures. Overall, the conclusion drawn in the manuscript are well supported by the presented results. However, the authors should provide a stronger rationale for their study, particularly in terms of the novelty and potential advancements in the field. The introduction section of the manuscript can be improved in order to entice the readers about the importance of the EPR measurements to evaluate the carbon film quality during the annealing process. Furthermore, the authors should make the manuscript read better. There are  grammatical errors in numerous sentences all over the manuscript. For example , line 36 on page 1, Electron paramagnetic resonance is defined as ESR which should be EPR. The authors should read the manuscript carefully before submission to avoid these errors. 

Response 1: Please provide your response for Point 1. (in red)

Electron paramagnetic resonance (EPR) is the most direct method of studying carbon films in condensed materials. The main direction in modern EPR studies is to increase the sensitivity and spectral resolution of spectrometers by increasing the operating frequency. Increasing the operating frequency also allows the registration of EPR spectra for systems with high spin states, including multi-particle paramagnetic complexes, significant fine structure splittings and exchange splittings of which, as a rule, do not allow the observation of EPR spectra at traditional frequencies. In this case, the possibility of lowering the operating temperature also plays a fundamental role, which allows, in addition to increasing the sensitivity, to determine the order of energy levels, i.e., to determine the signs of the fine structure parameters and exchange interactions.

The observation has been corrected. Although in the abstract it was stated. This paper presents the results of an electron paramagnetic resonance (EPR) study of carbon films deposited on quartz, mica, and silicon substrates.

Abstract: This paper presents the results of an electron paramagnetic resonance (EPR) study of carbon films deposited on quartz, mica, and silicon substrates.

Introduction

Recently, there has been considerable interest in the development of technologies for depositing thin carbon films on various materials.

The morphology, structure, phase composition, and physical properties of thin films directly depend on the method of their production. Various types of substrates are selected for obtaining such films, such as: quartz, glass, crystalline and monocrystalline silicon, etc.

 Depending on the synthesis temperature, the localized carbon can be presented both in an amorphous state and in a nanocrystalline state. The electron paramagnetic resonance (EPR) method is a powerful tool for studying carbon nanomaterials.

Point 2: The authors should certainly dedicate effort enhancing the sentence structure and thoroughly reviewing the grammar of the sentences presented in the manuscript.   

For example, here is one of the sentence in the manuscript: "Its phase differs from the influence of this field from the Super high frequency of the phase the signal of in the resonator itself." The meaning conveyed in this sentence is quite challenging to comprehend. The manuscript contains several sentences that are challenging to comprehend, making it difficult to discern the author's intended message. Breaking it into two shorter sentences could greatly aid in improving the clarity, explanation, and overall flow of information in the manuscript.Furthermore, I strongly recommend having a native English speaker review the final version of the manuscript, if possible.

Response 2: Please provide your response for Point 2. (in red)

In the resonator Super high frequency  the phase of the signal is different from the influence of this field. The observation has been corrected. The article has been translated.

Reviewer 4 Report

The objective of the study presented by the authors was to measure and determine the temperature dependence of the EPR signal in thin carbon films deposited on quartz, mica, and silicon substrates at different annealing temperatures. Overall, the conclusion drawn in the manuscript are well supported by the presented results. However, the authors should provide a stronger rationale for their study, particularly in terms of the novelty and potential advancements in the field. The introduction section of the manuscript can be improved in order to entice the readers about the importance of the EPR measurements to evaluate the carbon film quality during the annealing process. Furthermore, the authors should make the manuscript read better. There are  grammatical errors in numerous sentences all over the manuscript. For example , line 36 on page 1, Electron paramagnetic resonance is defined as ESR which should be EPR. The authors should read the manuscript carefully before submission to avoid these errors. 

The authors should certainly dedicate effort enhancing the sentence structure and thoroughly reviewing the grammar of the sentences presented in the manuscript.   

For example, here is one of the sentence in the manuscript: "Its phase differs from the influence of this field from the Super high frequency of the phase the signal of in the resonator itself." The meaning conveyed in this sentence is quite challenging to comprehend. The manuscript contains several sentences that are challenging to comprehend, making it difficult to discern the author's intended message. Breaking it into two shorter sentences could greatly aid in improving the clarity, explanation, and overall flow of information in the manuscript.

Furthermore, I strongly recommend having a native English speaker review the final version of the manuscript, if possible. 

Author Response

(The authors gave the same response as above.)

Reviewer 5 Report

Review Report

The manuscript “paramagnetic properties of carbon films” reports the electron paramagnetic resonance (ERP) study on carbon films deposited on various substrates including quartz, mica, and silicon. The ERP intensity, g-factor, and line width of these carbon films are found to be dependent on the temperature, which would help establish the relationship between carbon films’ properties and the preparation methods and shed light on the design and creation of functional materials. However, the current version of the manuscript was not well prepared, and it is challenging to grasp the authors’ ideas and even meanings, which consequently obscure the novelty of this work. Thus, I would advise the authors to re-organize and re-write the manuscript and then resubmit it for peer review. Authors are suggested to seek guidance about how to do scientific writing.

Below are some suggestions for improving the manuscript.

1.       A good Introduction is expected to provide the background, the state of the art, the gap of your research topic, as well as the main/key findings and the contributions of your work.

The current Introduction fails to make the readers understand the related information.

Also, the current manuscript fails to clearly state “Why are you studying the temperature dependence?”, “What kind of temperature dependences have you found?” and “Why do these dependencies exist?”

2.       The obvious errors should be avoided. For example, Lines 66-78 are repeated contents.

3.       The acronym EPR is suggested to be presented after the full name electron paramagnetic resonance, rather than before like in Lines 32-37.

Author Response

Response to Reviewer 5 Comments

Point 1: A good Introduction is expected to provide the background, the state of the art, the gap of your research topic, as well as the main/key findings and the contributions of your work.

The current Introduction fails to make the readers understand the related information.

Also, the current manuscript fails to clearly state “Why are you studying the temperature dependence?”, “What kind of temperature dependences have you found?” and “Why do these dependencies exist?” Below are some suggestions for improving the manuscript.

Response 1:

The synthesis of carbon films on different substrates was carried out from 100 0С to 800 0С. From a single sample the study is not obtained. The graphs were plotted by these parameters. The electron paramagnetic resonance (EPR) method is a powerful tool for studying carbon materials. Electron paramagnetic resonance is widely used to study carbon nanostructures and is the most direct method of studying carbon films in condensed materials. As the temperature rises to 750 0C, the normalized intensity of the EPR signal amplitude increases monotonically in all samples during carbon deposition on various substrates (quartz, mica, and silicon). EPR investigated the deposition of carbon on various substrates.

EPR measurement was carried out on 20 sprayed samples at temperatures of from 0 0C (initial) to of 800 0C.

Point 2: The obvious errors should be avoided. For example, Lines 66-78 are repeated contents.

Response 2:

I removed repeated lines in the article. The observation has been corrected.

Point 3: The acronym EPR is suggested to be presented after the full name electron paramagnetic resonance, rather than before like in Lines 32-37.

Response 3:

The observation has been corrected. Although in the abstract it was stated. This paper presents the results of an electron paramagnetic resonance (EPR) study of carbon films deposited on quartz, mica, and silicon substrates.

Abstract: This paper presents the results of an electron paramagnetic resonance (EPR) study of carbon films deposited on quartz, mica, and silicon substrates.

Introduction

Recently, there has been considerable interest in the development of technologies for depositing thin carbon films on various materials.

The morphology, structure, phase composition, and physical properties of thin films directly depend on the method of their production. Various types of substrates are selected for obtaining such films, such as: quartz, glass, crystalline and monocrystalline silicon, etc.

 Depending on the synthesis temperature, the localized carbon can be presented both in an amorphous state and in a nanocrystalline state. The electron paramagnetic resonance (EPR) method is a powerful tool for studying carbon nanomaterials.

Round 2

Reviewer 1 Report

In this version of the manuscript, the suggestions have been adequately taken into account and changes have been made accordingly, so it is suitable for publication for me.

Author Response

Point 1: See above for comments on English language.

 Response 1: The observation has been corrected.

Reviewer 4 Report

The authors have answered all the questions in the updated version of the manuscript. The English language has been improved significantly in the updated manuscript. There is a minor error in the sentences on page 2 line 90. The authors should remove "by the authors" from the sentence. 

Author Response

Point 1: The authors have answered all the questions in the updated version of the manuscript. The English language has been improved significantly in the updated manuscript. There is a minor error in the sentences on page 2 line 90. The authors should remove "by the authors" from the sentence. 

 Response 1: The observation has been corrected. Therefore, the line with g=2.0052 found of the article can be ascribed to the presence of graphite microcrystallites with some averaged g-factor in the deposited film

Reviewer 5 Report

It can be seen from this revision that authors have tried to improve the manuscript, with most errors being eliminated. But it is still challenging, or unpleasant to read this article and make a full understanding of it.  The data is sufficient to support the conclusion, but the data discussion and the narrative (or the scientific writing) still have a big room for improvement. For example,

1. The meaning of this study is not clearly presented in Introduction. 

2. Why the similar EPR signal evolutions of parallel and perpendicular samples with temperature can confirm the same paramagnetic center which is characterized by g at 750 C?

3. Line 237-241 cannot explain the relationship between the EPR line width (the data) and factors such as microwave power, temperature, and magnetic field amplitude. 

Minor errors

1. Line 186 "2.00118 ./. 2.00164"?

2. The caption of Figure 2 is not complete.

3. Figure 3a, the labels for perpendicular magnetic field are wrong.

4. Line 220, "Temperature growth of the EPR signals" might be "EPR signal growth with temperature".

Author Response

Response to Reviewer 5 Comments

Point 1: The meaning of this study is not clearly presented in Introduction. 

Response 1:

The purpose of this work is to study the study of carbon films deposited on substrates of quartz, mica, and silicon by electron paramagnetic resonance (EPR). In this paper the results of electron paramagnetic resonance spectroscopy were used.

Point 2: Why the similar EPR signal evolutions of parallel and perpendicular samples with temperature can confirm the same paramagnetic center which is characterized by g at 750 C?

Response 2: It turned out that the temperature behavior of the EPR signal at the perpendicular and parallel arrangement of the sample plane relative to the orientation of the magnetic field is almost similar.  This means that the EPR signal refers to the same paramagnetic center, characterized by g=2.00322-2.00323 at T=750 0С. The temperature T=750 0C shows that the optimal deposition and formation of carbon films and g-factor are the same.

Point 3:

  Line 237-241 cannot explain the relationship between the EPR line width (the data) and factors such as microwave power, temperature, and magnetic field amplitude. 

Response 3: This means that the effect of the Super High Frequency power and the magnetic field modulation amplitude on the behavior of the EPR signals appears to be the same. This indicates that the centers have identical relaxation times, which may be related to the same local environment of the centers, or to the same dynamic properties [21-23].

Electron paramagnetic resonance (EPR) is the phenomenon of resonant absorption of electromagnetic radiation by a paramagnetic substance placed in a constant magnetic field. It is caused by quantum transitions between magnetic sublevels of paramagnetic atoms and ions (Zeeman effect). EPR spectra are observed mainly in the Super High Frequency. The electron paramagnetic resonance method allows us to evaluate the effects manifested in EPR spectra due to the presence of local magnetic fields. In turn, local magnetic fields reflect the pattern of magnetic interactions in the system under study. Thus, the method of EPR spectroscopy allows us to study both the structure of paramagnetic particles and the interaction of paramagnetic particles with the environment.

Peculiarities of the EPR spectrometer operation

Paramagnetic sample in a special cuvette (ampoule or capillary) is placed inside the working resonator located between the poles of the electromagnet of the spectrometer. Electromagnetic microwave radiation of constant frequency enters the resonator. The resonance condition is achieved by linearly varying the magnetic field strength. High-frequency modulation of the magnetic field is used to increase the sensitivity and resolution of the analyzer.

Minor errors

  1. Line 186 "2.00118 ./. 2.00164"?

Response 1. The observation has been corrected. The third component of the carbon film has a g-factor equal to 2.00118 - 2.00164 depend on the angle of rotation of the sample in a magnetic field.

  1. The caption of Figure 2 is not complete.

Initial data of carbon film on mica substrate at 4000C under magnetic field. Measurements of the perpendicular location of the sample plane relative to the orientation of the magnetic field: 1-2 - Mn signal, 3-8 - sample signals and EPR characteristics, 6 - center signal. The observation has been corrected.

  1. Figure 3a, the labels for perpendicular magnetic field are wrong.

Response 3. The observation has been corrected.

  1. Line 220, "Temperature growth of the EPR signals" might be "EPR signal growth with temperature".

Response 4. The observation has been corrected. EPR signal growth with temperature in the studied samples were observed……

Response 4. The observation has been corrected. EPR signal growth with temperature in the studied samples were observed in the interval T= 400–700 0С when measured in the parallel position of the sample plane relative to the direction of the magnetic field (figure 3b).
